# Risk and Protective Factors for Frailty in Pre-Frail and Frail Older Adults

**DOI:** 10.3390/ijerph20043123

**Published:** 2023-02-10

**Authors:** Juan Corral-Pérez, Laura Ávila-Cabeza-de-Vaca, Andrea González-Mariscal, Milagrosa Espinar-Toledo, Jesús G. Ponce-González, Cristina Casals, María Ángeles Vázquez-Sánchez

**Affiliations:** 1ExPhy Research Group, Department of Physical Education, University of Cadiz, Puerto Real, 11519 Cadiz, Spain; 2Biomedical Research and Innovation Institute of Cadiz (INiBICA) Research Unit, Puerta del Mar University Hospital, 11009 Cadiz, Spain; 3Clinical Management Unit, Malaga-Guadalhorce Health District, Rincón de la Victoria, 29730 Malaga, Spain; 4Department of Nursing, Faculty of Health Sciences, University of Malaga, 29071 Malaga, Spain; 5PASOS Research Group, UMA REDIAS Network of Law and Artificial Intelligence Applied to Health and Biotechnology, University of Malaga, 29071 Malaga, Spain

**Keywords:** physical function, physical activity, accelerometry, dependency, strength, ageing

## Abstract

This study aims to evaluate the differences in body composition, physical function, and physical activity between pre-frail/frail older adults and to detect risk and protective factors against frailty and physical frailty. Fried’s criteria for frailty and physical frailty using the short-performance physical battery (SPPB) were measured in 179 older participants (75.3 ± 6.4 years old). Body weight, height, and waist, arm, and leg circumferences were obtained as body composition variables. Daily accelerometer outcomes (physical activity and inactivity) were obtained. Pre-frail participants showed overall better physical function and spent more time in physical activity and less time in long inactivity periods than frail participants (*p* < 0.05). Risk frailty factors were higher waist perimeter (Odds Ratio [OR]: 1.032, 95%CI: 1.003–1.062), low leg performance (OR: 1.025, 95%CI: 1.008–1.043), and inactivity periods longer than 30 min (OR: 1.002, 95%CI: 1.000–1.005). Protective factors were standing balance (OR: 0.908, 95%CI: 0.831–0.992) and SPPB score (OR: 0.908, 95%CI: 0.831–0.992) for frailty, handgrip strength (OR: 0.902, 95%CI: 0.844–0.964) for physical frailty, and light (OR: 0.986, 95%CI: 0.976–0.996) and moderate-to-vigorous (OR: 0.983, 95%CI: 0.972–0.996) physical activity for both. Our findings suggest that handgrip strength, balance, and physical activity are protective frailty factors and can be monitored in pre-frail older adults. Moreover, poor lower body performance and long inactivity periods are frailty risk factors, which highlights their importance in frailty assessment.

## 1. Introduction

Frailty has been defined as a progressive age-related decline characterized by physiological and functional dysfunction that leads to a state of vulnerability and lack of reserve to stressors [1]. Frailty has been described as a continuum state constituted by two phases, pre-frailty and frailty [2]. On one hand, the adverse outcomes of pre-frailty are not as high as those in frailty, this phase has been also associated with an increased risk of falls and mortality [1]. On the other hand, a person who is considered frail is more prone to adverse health outcomes like fallings, dependency, fractures, hospitalization, and death [3].

Frailty has become a common state in the elder population due to the increase in life expectancy [4], with an estimated prevalence in Europe of around 50% of the population suffering from either pre-frailty or frailty [5]. Consequently, frailty has become one of the most researched topics in the current literature. Therefore, the importance of an early diagnosis (e.g., pre-frailty) is highlighted, as well as the analysis of risk and protective factors for frailty to prevent the serious consequences of frailty [6].

However, several scales to detect frailty are currently being used [7]. In this sense, the Fried scale [1] is one of the most evaluated and frequently used scales [7], and provides five phenotypes of frailty. Further, as impaired physical function is a major indicator of frailty, the current literature highlights the Short Performance Physical Battery (SPPB) [8] as one of the best physical performance tests to identify frail older adults. Accordingly, both scales are included in the present study.

The main factors that can modulate physical fitness in frailty are physical activity and inactivity, being of high interest regarding their analysis as protective or risk factors for frailty [9,10]. Thus, whereas physical activity has been proposed as a fundamental strategy to prevent the appearance of frailty [11], it is still less clear how bouted physical activity is related to frailty. Similarly, the associations between inactivity and frailty are inconsistent, probably due to most of the studies studying total inactivity time regardless of the accumulation pattern [12].

For all these reasons, the present study aims to compare the differences between pre-frail and frail older people in several key aspects of everyday life, such as physical fitness, anthropometric measures, and daily physical activity and inactivity (both in total time and bouted), and to evaluate the associations with Fried’s frailty criteria and physical frailty.

## 2. Materials and Methods

### 2.1. Design and Participants

A multicentre cross-sectional study was conducted in the provinces of Málaga and Cádiz (Spain) between January and July 2022. Participants were included if they were 65 years old or older, presented at least one condition of Fried’s frailty phenotype, and were able to travel alone or accompanied to the evaluations. Older adults with dementia or who were institutionalized were excluded.

When participants were interested in participating, the information sheet and the informed consent were provided, explaining all the procedures and potential risks associated with the study. Both documents were returned signed before the start of the study. All procedures were approved by the Ethics Committee of the Provincial Research of Málaga (reference FRAGSALUD, date 31 January 2019) and followed the Declaration of Helsinki for human studies.

### 2.2. Frailty Assessment

The frailty status of all participants was measured using the Fried’s frailty phenotype criteria, which classifies older adults as robust, pre-frail, or frail [1] based on their final score using the following domains: (i) unintentional weight loss, (ii) self-reported exhaustion, (iii) low weekly physical activity expenditure through the Minnesota Leisure Time Activity Questionnaire), (iv) low gait speed in 4.57 gait test, and lastly (v) low handgrip strength. Participants were classified as pre-frail if they met one or two of the criteria, and as frail if they met three or more criteria.

### 2.3. Physical Frailty

Physical frailty was measured using SPPB [8], a reliable and valid measure of physical performance in older adults (intraclass correlation coefficient ranging 0.82–0.92), although it is not particularly sensitive to change (ranging 0.7–3.42) [13]. This method classifies older adults depending on their performance in 3 physical assessments: (i) gait speed using a 4-m walk test, (ii) lower limb strength using the five-repetition sit-to-stand test, and (iii) balance through performing three tests (side-by-side, semi-tandem, and tandem stands). Participants’ performance in each test was compared to their normative data and scored between 0 and 4 points. If the participants could not perform the test, their score was reported as 0. In the end, participants obtained a score from 0 (highly dependent) to 12 (totally independent).

A score lower than 9 points is considered to be classified as frail [8], therefore participants were divided into physically pre-frail (SPPB scores between 9 and 12), and physically frail (SPPB scores lower than 9).

### 2.4. Body Composition

Body weight was obtained using a digital scale (Omron Medizintechnik, Mannheim, Germany). Body height was measured in a standing position on the Frankfort plane, after normal expiration with a stature-measuring instrument (SECA 225, Hamburg, Germany). From both variables, body mass index (BMI) was calculated using the formula weight(kg)/height(m)^2^. The waist, arm, and leg circumferences (cm) were assessed with a metallic non-extensible tape (Lufkin W606PM, Washington, DC, USA) at the level of the thinnest part of the waist between the iliac crest and the last rib, and the longest circumference of the arm and leg, respectively.

### 2.5. Assessment of Physical Activity and Inactivity Time

Daily physical activity and inactivity time were measured using a wrist-worn accelerometer (GENEActiv, ActivInsights Ltd., Kimbolton, UK). Participants wore the accelerometer on their non-dominant wrist for a minimum of 6 consecutive days. Only results from participants with wear time ≥ 16 h/d during at least 4 days (at least 3 weekdays and at least 1 weekend day) were considered valid.

Accelerometers were set at 60 Hz and raw data were downloaded using GENEActiv software (version 3.3). All raw data files were managed on the University of Malaga servers and consequently analysed by the open-source R-package GGIR, version 2.5-0 (https://cran.r-project.org/web/packages/GGIR/index.html access on 6 December 2022) using R-package (R Core Team, Vienna, Austria). This R-package was used to minimize the sensor calibration error (auto-calibration of the data based on local gravity) [14] and accelerations determined by calculating the Euclidean norm minus one (ENMO). Four main categories according to the intensity were established according to ENMO cut-points developed for the non-dominant wrist [15], including inactivity (<1.5 Metabolic Equivalents (METs) for intensities <40 mg), Light Physical Activity (LPA) (1.5–2.99 METs for intensities ≥40 mg and <100 mg), and Moderate-to-Vigorous Physical Activity (MVPA) (>3 METs for intensities ≥ 100 mg).

Regarding inactivity variables, total daily inactivity time, and daily inactivity time in 10–20 min bouts, 20–30 min bouts, and >30-min bouts were calculated. Concerning physical activity, total daily LPA time, LPA time in 1–10 min, and >10-min bouts, total daily MVPA, and MVPA in bouts of 1–5, 5–10, and >10 min were calculated.

### 2.6. Statistical Analyses

All data are expressed as mean ± standard deviation (SD). Normality (Kolmogorov-Smirnov) and homogeneity of variance (Levene test) tests were performed.

To compare the differences on physical fitness, body composition, physical activity, and inactivity between pre-frail and frail participants based on Fried’s criteria [1], a one-way analysis of variance (ANOVA) was performed including age and sex as covariables.

Logistic regressions were used to explore the associations between Fried’s frailty criteria with physical fitness, body composition, physical activity, and inactivity, without the variables included in the criteria. Similar to this, logistic regressions were also used to study the associations between SPPB physical frailty and the above-mentioned variables, excluding again those included in their evaluation. In both logistic regression analyses, sex and age were included in the adjusted models as covariables.

Lastly, linear regression analyses (step-wise method) were performed to analyse the associations between SPPB final score and physical fitness, body composition, physical activity, and inactivity. Additionally, we fitted the models over the principal components extracted using principal component analyses (PCA) following the parallel analyses to determine the models [16]. The PCA reported the following four clusters (Figure A1): RC1 which includes inactivity and LPA, RC2 which includes MVPA, RC3 which includes physical tests, and RC4 which includes all body composition variables (BMI, waist, arm, and leg circumferences). The final model included all significant variables of the previous four models. All models were adjusted by age and sex.

All analyses were performed by using the IBM SPSS Statistics 26 software (SPSS Inc., Chicago, IL, USA), with a significance set at *p* < 0.05.

## 3. Results

### 3.1. Pre-Frailty and Frailty Differences

A total of 179 older adults (113 females) were included in the study. Using Fried’s criteria, 68% of participants were classified as pre-frail while the rest were included in the frail group. A comparison between these groups showed significant differences in most of the variables (Table 1).

### 3.2. Associations between Fried’s Frailty Criteria and Physical Fitness, Anthropometric Measures, and Daily Physical Activity and Inactivity

We examined the associations of Fried’s frailty criteria groups (pre-frail and frail groups) with physical fitness, anthropometric measures, and daily physical activity and inactivity in the logistic regression analyses (Table 2). Among the 20 variables included in this analysis, 3 of them significantly increases the risk of transitioning from pre-frailty to frailty (waist perimeter, the time in completing the sit-to-stand test, and the inactivity time accumulated bouts longer than 30 min), while 9 were associated as protective risk factors against the development of frailty (Stand test, SPPB score, total LPA, LPA in bouts between 1 and 10 min, total MVPA and inactivity bouts between 1 and 10 min).

### 3.3. Associations between Physical Frailty and Physical Fitness, Anthropometric Measures, and Daily Physical Activity and Inactivity

We examined the associations of SPPB physical frailty groups with physical fitness, anthropometric measures, and daily physical activity and inactivity in the logistic regression analyses. In this model, only handgrip strength, LPA in bouts between 1 and 10 min, total MVPA, and inactivity bouts between 10 and 20 min and between 1 and 10 min were significantly associated as protective factors against physical frailty (Table 3).

The associations between SPPB final score and physical fitness, anthropometric measures, and daily physical activity and inactivity are shown in Table 4. Body composition variables were not associated with SPPB final score. Handgrip strength, age, and total MVPA were the main predictors of the SPPB final score accounting for 27.1%.

## 4. Discussion

The data from this study showed that pre-frail participants had a lower waist perimeter, an overall better physical function than frail participants, and higher levels of LPA and MVPA than frail older adults, whereas frail participants accumulated more inactive time periods longer than 30 min. In addition to this, waist perimeter, the time of performing sit to stand test, and longer inactivity bouts were associated with increased risk of Fried’s frailty while higher levels of balance, LPA and MVPA, and short inactivity bouts were associated as protective factors in the transition from pre-frailty to frailty. Handgrip strength, LPA and MVPA, and shorter inactivity bouts were associated as protective factors against physical frailty, with handgrip strength and MVPA levels being the best predictors of physical independence.

Physical frailty is a significant risk factor for negative health outcomes [17]. Our data showed that pre-frail participants had an overall better physical performance than their frail counterparts. Regarding physical function, it has been previously shown that non-frail elders had higher mean handgrip values, better performance in lower body strength, better balance, and faster walking speed than frail elders [9]. Our results displayed that these differences were still present during the process from pre-frailty to frailty, with pre-frail participants performing higher values of handgrip strength, standing balance, lower body performance, and overall SPPB scores. However, despite the fact that the frailty status seems to have power to predict adverse health outcomes, such as falls [18], we did not detect any significant differences in more demanding balance tests like the semi-tandem and tandem tests, suggesting that the decrease in these static balance tests is significant from the moment a person can be considered pre-frail.

Concerning physical activity and inactivity, pre-frail older adults accumulated more time in LPA and MVPA, whereas frail older adults spent more time in inactivity periods longer than half an hour. These results are similar to what other authors have found recently, with frail adults spending less time doing physical activity and accumulating more time in inactive behaviour [19]. However, our results presented an interesting outcome. Frail participants spent more inactivity time than pre-frail older adults, but with only a trend to signification (*p* = 0.066). However, there were differences in inactive time when it was fragmented into different bouts. Older adults with frailty tended to accumulate more inactive time in longer periods (>30 min), whereas pre-frail older adults accumulate more inactive time in shorter periods (<30 min). These results could suggest that the adverse effects of inactivity on frailty appear when this behaviour is maintained during prolonged periods of time, similar to the findings of other authors who showed longer periods of inactivity time without any physical activity breaks are also related to diminished physical function in older adults [20].

Higher levels of waist perimeter as well as poorer performance in the sit-to-stand test, which could be considered as a marker of lower body performance, were associated as independent risk factors of frailty based on Fried’s criteria [1]. A higher waist perimeter, as a marker of central obesity, has been proposed as a frailty risk in previous studies [5]. This association might be explained by the increased release of pro-inflammatory markers by the adipocytes that contribute to inflammation, which has been related to increased frailty in older adults [21]. The sit-to-stand test has also been associated with a higher risk of suffering from frailty [22]. This association may be explained by the suggested mediator role of lower body performance in maintaining a good physical status, since reduced lower body performance has been associated with lower physical performance and all-cause mortality in older adults [23]. This could suggest that the activation and use of the lower body muscles stimulate the release of several markers, such as myokines, that help to maintain the muscle mass quality and function of older adults [24].

Nonetheless, this study also showed protective factors against frailty such as the side-by-side test and SPPB scores. The association between balance and frailty might be explained by the role of static balance in reducing the risk of falls, with interventional programs focused on the improvement of static balance showing a reduction between 22–34% of falls in older adults [25]. SPPB scores are an indicator of physical frailty [8], which indicates that an overall better physical function is a protective factor against the development of frailty. In line with this, in this study, handgrip strength has been associated as a protective factor against physical frailty. Handgrip strength has been associated with independency, with previous studies that showed lower values of handgrip being associated with reduced independence and mobility in men and women [26]. Therefore, handgrip strength could be an effective diagnostic tool for frailty, regardless of the protocol used.

Our results showed that LPA is a protective factor against both the development of frailty based on Fried’s criteria [1] and physical frailty based on SPPB scores [8]. LPA, especially higher levels, has also been associated with frailty-related morbidities and disability regardless of the time spent in MVPA [27], reduced levels of mortality, and cardiometabolic risk, as well as reduced frailty risk [11,27]. Furthermore, the association between LPA and frailty might also explain the positive association between short inactivity periods (between 1 and 10 min) since older adults who usually break their inactivity behaviour might accumulate more minutes doing LPA periods (especially between 1 and 10 min), following the World Health Organization recommendations [28], and consequently reducing their risk frailty. This can also explain why shorter inactivity bouts were associated as protective factors for both measurements.

Similar to LPA, MVPA had been associated as a protective factor against both frailty and physical frailty. The association between MVPA and frailty may be explained by the intensity of the exercise, which is suggested to promote an improvement in muscle protein synthesis through the higher acute production of reactive oxygen species and pro-inflammatory cytokines [29], which lead to muscular adaptations that promote the maintaining of muscle mass [30] and muscle function [31]. Nonetheless, given the multiple linear regressions, MVPA showed to be a better predictor of physical function than LPA, suggesting that increasing the levels of MVPA could be a more efficient strategy to reduce physical frailty in older adults.

Our study also has limitations. The cross-sectional design of this study does not support definitive conclusions, and further studies are encouraged to determine the impact of the factors associated in this study with frailty. Additionally, to this, the accelerometer was worn on the wrist, which did not allow us to differentiate sedentary behaviour. Nonetheless, the present study also has strengths, including two of the most used frailty criteria used in the literature [1,8]. In addition to this, the wrist-worn accelerometer is waterproof, which allows us to estimate the physical activity and inactivity during the whole day, including water-based activities, avoiding the possible loss of data due to the removal of the accelerometer for these activities.

## 5. Conclusions

The results of this study showed that pre-frail older adults had better physical function, higher levels of physical activity, and lower levels of prolonged inactivity behaviour than their frail counterparts.

Our findings suggest that a higher waist perimeter, a poor lower body performance, and long inactivity periods are frailty risk factors, which highlights their importance in frailty assessment. Moreover, handgrip strength, balance, and physical activity are protective frailty factors.

Therefore, in an increasingly aging population, the mentioned risk and protective factors should be monitored in pre-frail older adults in order to prevent or delay frailty-related worsening and to decrease negative health outcomes, such as falls or all-cause mortality.

## Figures and Tables

**Table 1 ijerph-20-03123-t001:** Participant characteristics by Fried’s frailty phenotype.

	Total (n = 179)	Pre-Frail (n = 122)	Frail (n = 57)	*p*
Sex, n (%)				**0.026**
Men	66 (36.9)	49 (42.1)	17 (24.5)	
Women	113 (63.1)	73 (57.9)	40 (75.5)	
Age (years)	75.31 ± 6.40	74.52 ± 5.77	77.21 ± 7.41	**0.008**
BMI (kg/m^2^)	29.19 ± 4.80	29.19 ± 4.77	29.20 ± 4.91	0.782
Waist perimeter (cm)	99.47 ± 11.62	98.62 ± 11.63	101.55 ± 13.81	**0.029**
Arm Perimeter (cm)	28.90 ± 3.80	29.00 ± 3.54	28.64 ± 4.41	0.713
Calf perimeter (cm)	35.14 ± 4.41	35.70 ± 4.41	33.70 ± 4.12	0.078
Physical activity expenditure (kcal/week)	3131.44 ± 3222.38	3596.63 ± 3361.62	2086.96 ± 2625.99	**0.018**
Handgrip strength (kg)	21.59 ± 9.89	24.01 ± 9.61	16.26 ± 8.35	**<0.001**
Side-by-side Test (s)	8.34 ± 3.05	8.78 ± 3.05	7.27 ± 4.36	**0.027**
Semi-tandem test (s)	7.85 ± 3.74	8.14 ± 3.50	7.15 ± 4.24	0.300
Tandem test (s)	6.08 ± 4.16	6.47 ± 4.05	5.14 ± 4.31	0.239
4-m Gait test (s)	5.39 ± 2.22	4.87 ± 1.28	6.68 ± 3.28	**<0.001**
Sit-to-Stand Test (s)	22.31 ± 18.18	19.25 ± 15.32	29.81 ± 22.21	**0.001**
SPPB score	7.83 ± 2.54	8.55 ± 2.13	6.08 ± 2.63	**<0.001**
Total LPA (min/day)	198.95 ± 89.69	209.35 ± 86.15	174.12 ± 93.93	**0.026**
LPA bouts >10 min (min/day)	83.59 ± 59.51	87.57 ± 59.44	74.09 ± 59.19	0.152
LPA bouts of 1–10 min (min/day)	115.36 ± 39.38	121.78 ± 36.30	100.02 ± 42.50	**0.008**
Total MVPA (min/day)	39.95 ± 39.35	45.96 ± 40.23	25.65 ± 33.40	**0.006**
MVPA bouts >10 min (min/day)	10.07 ± 17.35	12.65 ± 18.99	3.90 ± 10.39	**0.020**
MVPA bouts of 5–10 min (min/day)	12.29 ± 13.17	14.17 ± 13.36	7.78 ± 11.64	**0.014**
MVPA bouts of 1–5 min (min/day)	17.86 ± 14.47	19.93 ± 14.48	12.91 ± 13.32	**0.014**
Total inactivity (min/day)	716.68 ± 128.54	703.86 ± 128.60	747.29 ± 124.38	0.066
Inactivity bouts >30 min (min/day)	524.61 ± 171.62	502.73 ± 166.37	576.85 ± 174.33	**0.019**
Inactivity bouts of 20–30 min (min/day)	56.51 ± 21.80	58.23 ± 21.04	52.41 ± 23.25	0.185
Inactivity bouts of 10–20 min (min/day)	60.98 ± 24.62	63.41 ± 23.57	55.10 ± 26.31	0.114
Inactivity bouts of 1–10 min (min/day)	88.39 ± 30.24	93.21 ± 28.29	76.87 ± 31.88	**0.002**

Values are expressed as mean ± standard deviation, and significant differences are highlighted in bold. P, *p* value; BMI, Body Mass Index; SPPB, Short-performance physical battery; LPA, Light Physical Activity; MVPA, Moderate to vigorous physical activity.

**Table 2 ijerph-20-03123-t002:** Logistic regression analysis of the association between possible risk factors and Fried’s frailty.

	Frail OR (95% CI)	*p*	Adjusted Frail OR (95% CI)	*p*
BMI (kg/m^2^)	1.023 (0.963–1.087)	0.464	1.011(0.943–1.084)	0.761
Waist perimeter (cm)	1.021 (0.996–1.048)	0.103	**1.032 (1.003–1.062)**	**0.033**
Arm Perimeter (cm)	0.966 (0.892–1.045)	0.385	0.980 (0.895–1.074)	0.670
Calf perimeter (cm)	**0.921 (0.854–0.994)**	**0.034**	0.927 (0.851–1.010)	0.082
Side-by-side Test (s)	**0.886 (0.817–0.961)**	**0.004**	**0.908 (0.831–0.992)**	**0.033**
Semi-tandem test (s)	**0.923 (0.854–0.998)**	**0.044**	0.957 (0.878–1.042)	0.311
Tandem test (s)	**0.916 (0.850–0.987)**	**0.022**	0.953 (0.878–1.034)	0.245
Sit-to-Stand Test (s)	**1.029 (1.013–1.045)**	**<0.001**	**1.025 (1.008–1.043)**	**0.005**
SPPB score	**0.639 (0.547–0.745)**	**<0.001**	**0.670 (0.566–0.793)**	**<0.001**
Total LPA (min/day)	**0.996 (0.992–1.000)**	**0.030**	**0.995 (0.991–1.000)**	**0.034**
LPA bouts >10 min (min/day)	0.997 (0.991–1.002)	0.237	0.996 (0.989–1.002)	0.176
LPA bouts of 1–10 min (min/day)	**0.986 (0.978–0.955)**	**0.002**	**0.986 (0.976–0.996)**	**0.005**
Total MVPA (min/day)	**0.983 (0.972–0.994)**	**0.002**	**0.983 (0.972–0.996)**	**0.007**
MVPA bouts >10 min (min/day)	**0.953 (0.919–0.987)**	**0.008**	**0.954 (0.917–0.994)**	**0.023**
MVPA bouts of 5–10 min (min/day)	**0.957 (0.927–0.988)**	**0.008**	**0.959 (0.926–0.992)**	**0.017**
MVPA bouts of 1–5 min (min/day)	**0.963 (0.938–0.989)**	**0.006**	**0.965 (0.938–0.994)**	**0.017**
Total inactivity (min/day)	1.002 (1.000–1.005)	0.056	1.003 (1.000–1.005)	0.071
Inactivity bouts >30 min (min/day)	**1.002 (1.000–1.004)**	**0.012**	**1.002 (1.000–1.005)**	**0.023**
Inactivity bouts of 20–30 min (min/day)	0.985 (0.970–1.000)	0.052	0.989 (0.973–1.005)	0.184
Inactivity bouts of 10–20 min (min/day)	**0.984 (0.971–0.998)**	**0.022**	0.988 (0.974–1.003)	0.124
Inactivity bouts of 1–10 min (min/day)	**0.983 (0.971–0.994)**	**0.003**	**0.980 (0.967–0.994)**	**0.004**

Adjusted OR ratio included sex and age as covariables. Significant differences are highlighted in bold. OR, Odd Ratio; 95% CI, 95% Confidence Interval, BMI, Body Mass Index; SPPB, Short-performance physical battery; LPA, Light Physical Activity; MVPA, Moderate to vigorous physical activity.

**Table 3 ijerph-20-03123-t003:** Logistic regression analysis of the association between possible risk factors and SPPB’s physical frailty.

	Frail OR (95% CI)	*p*	Adjusted Frail OR (95% CI)	*p*
BMI (kg/m^2^)	1.006 (0.950–1.064)	0.840	1.002(0.938–1.069)	0.960
Waist perimeter (cm)	0.997 (0.974–1.020)	0.769	1.007 (0.981–1.035)	0.590
Arm Perimeter (cm)	0.978 (0.911–1.050)	0.543	0.982 (0.903–1.068)	0.668
Calf perimeter (cm)	0.952 (0.891–1.016)	0.139	0.976 (0.903–1.054)	0.534
Physical activity expenditure (kcal/week)	**1.000 (1.000–1.000)**	**0.026**	1.000 (1.000–1.000)	0.232
Handgrip strength (kg)	**0.921 (0.889–0.954)**	**<0.001**	**0.927(0.882–0.974)**	**0.003**
Total LPA (min/day)	0.998 (0.995–1.001)	0.202	0.997(0.993–1.001)	0.100
LPA bouts >10 min (min/day)	0.999(0.994–1.004)	0.682	0.997(0.991–1.003)	0.301
LPA bouts of 1–10 min (min/day)	**0.991 (0.983–0.999)**	**0.020**	**0.989 (0.980–0.998)**	**0.023**
Total MVPA (min/day)	**0.989 (0.981–0.997)**	**0.009**	**0.992(0.983–0.998)**	**0.048**
MVPA bouts >10 min (min/day)	**0.972 (0.952–0.992)**	**0.006**	0.980 (0.958–1.002)	0.071
MVPA bouts of 5–10 min (min/day)	0.977 (0.954–1.001)	0.056	0.986 (0.960–1.012)	0.288
MVPA bouts of 1–5 min (min/day)	**0.977 (0.986–0.998)**	**0.031**	0.980 (0.956–1.005)	0.117
Total inactivity (min/day)	1.000 (0.998–1.003)	0.697	1.001 (0.998–1.003)	0.670
Inactivity bouts >30 min (min/day)	0.992 (0.979–1.006)	0.249	1.001 (0.999–1.003)	0.232
Inactivity bouts of 20–30 min (min/day)	0.985 (0.970–1.000)	0.249	0.989 (0.973–1.005)	0.367
Inactivity bouts of 10–20 min (min/day)	**0.981 (0.969–0.994)**	**0.005**	**0.980 (0.966–0.995)**	**0.011**
Inactivity bouts of 1–10 min (min/day)	0.992 (0.982–1.001)	0.092	**0.986 (0.974–0.999)**	**0.028**

Adjusted OR ratio included sex and age as covariables. Significant differences are highlighted in bold. OR, Odd Ratio; 95% CI, 95% Confidence Interval BMI, Body Mass Index; LPA, Light Physical Activity; MVPA, Moderate to vigorous physical activity.

**Table 4 ijerph-20-03123-t004:** Regression analyses of SPPB scores and body composition, physical function, and physical activity in pre-frail/frail older adults.

	B	SE B	β	*p*
Model RC1				
Age (years)	−0.015	0.027	−0.280	**<0.001**
Sex (0 = men, 1 = women)	−1.095	0.366	−0.212	**0.003**
Model RC2				
Age (years)	−0.103	0.028	−0.276	**<0.001**
Sex (0 = men, 1 = women)	−1.057	0.369	−0.205	**0.001**
Total MVPA (min/day)	0.011	0.005	0.176	**0.022**
Model RC3				
Handgrip strength (kg)	0.083	0.017	0.329	**<0.001**
Physical activity expenditure (kcal/week)	0.000	0.000	0.223	**0.001**
Age (years)	−0.083	0.027	−0.216	**0.002**
Model RC4				
Age (years)	−0.132	0.270	−0.350	**<0.001**
Sex (0 = men, 1 = women)	−0.923	0.352	−0.187	**0.009**
Total MVPA (min/day)	0.010	0.005	0.167	**0.030**
Final model				
Handgrip strength (kg)	0.091	0.018	0.38	**<0.001**
Age (years)	−0.065	0.029	−0.179	**0.024**
Total MVPA (min/day)	0.010	0.005	0.167	**0.030**

Model RC1-Dependent variable: SPPB score, R^2^ = 0.450, AdjR^2^ = 0.202, *p* = 0.004. Excluded variables: Sex, Total inactivity (min/day), Inactivity bouts >30 min (min/day), Inactivity bouts of 20–30 min (min/day). Model RC2-Dependent variable: SPPB score, R^2^ = 0.186, AdjR^2^ = 0.171, *p* = 0.022. Excluded variables: MVPA bouts >10 min (min/day), MVPA bouts of 5–10 min (min/day), MVPA bouts of 1–5 min (min/day). Model RC3-Dependent variable: SPPB score, R^2^ = 0.284, AdjR^2^ = 0.271, *p* = 0.002. Excluded variables: Sex. Model RC4-Dependent variable: SPPB score, R^2^ = 0.157, AdjR^2^ = 0.147, *p* = 0.009. Excluded variables: Weight, Body Mass Index, Waist circumference, arm circumference and leg circumference. Final Model-Dependent variable: SPPB score, R^2^ = 0.279, AdjR^2^ = 0.264, *p* = 0.030. Excluded variables: Sex, physical activity expenditure, Inactivity bouts of 10–20 min (min/day). Significant differences are highlighted in bold.

## Data Availability

The data presented in this study are available on reasonable request from the corresponding author.

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
