# Peer review of "Risk and Protective Factors for Frailty in Pre-Frail and Frail Older Adults"

_ijerph, 2023, doi:10.3390/ijerph20043123_

Round 1

Reviewer 1 Report

Dear authors,

It has been a pleasure to read your article, I found it very interesting. However, I would like to add a few comments:

Line 46 à To put examples inside the parentheses use 'e.g.'

Line 54 à Although I understand more or less where you want to go with this reasoning, I encourage you to rewrite this part in order to improve the justification of the study.

Line 71 à The number of participants is usually located in the results section.

Line 81 à Is that the number or code assigned by the ethics committee?

Line 92 à When possible, it could indicate the psychometric data of the variables.

Line 140 à Why do you use an ANOVA test to compare both groups?

Line 158 à Please include here the number of participants of the study.

Line 162 à Table 1. Please indicate the meaning of ‘p’ at the end of the table.

Author Response

STATUS: Pending minor revisions

Dear authors,

It has been a pleasure to read your article, I found it very interesting. However, I would like to add a few comments:

Point 1: Line 46 à To put examples inside the parentheses use 'e.g.'

Response 1: Corrected after suggestion. Thank you very much for you comments.

Point 2: Line 54 à Although I understand more or less where you want to go with this reasoning, I encourage you to rewrite this part in order to improve the justification of the study.

Response 2: We have rewritten the sentence and highlighted in yellow in the revised manuscript, as follows: “The main factors that can modulate physical fitness in frailty are physical activity and inactivity, being of high interest its analysis as protective or risk factors for frailty”.

Point 3: Line 71 à The number of participants is usually located in the results section.

Response 3: Corrected after suggestion.

Point 4: Line 81 à Is that the number or code assigned by the ethics committee?

Response 4: Yes, it is. Our code is not a number as we have seen in other Ethics Committees, our code is FRAGSALUD.

Point 5: Line 92 à When possible, it could indicate the psychometric data of the variables.

Response 5: Thank you very much. We have included information of validity, reliability and sensitivity.

Point 6: Line 140 à Why do you use an ANOVA test to compare both groups?

Response 6: Because it allows to include age and sex as covariables. Notwithstanding, the Student t test for independent samples showed significant differences, but as sex and age differed between groups, we think that is more appropriated to perform the ANOVA with both covariables.

Point 7: Line 158 à Please include here the number of participants of the study.

Response 7: Done. Thank you.

Point 8: Line 162 à Table 1. Please indicate the meaning of ‘p’ at the end of the table.

Response 8: It has been included after suggestion.

Reviewer 2 Report

Summary: Although the summary is correct, the mean and deviation of the age of the participants should be included.

Introduction and theoretical framework

- The theoretical framework presents the main orientations and references so that the theoretical background of the paper can be presented and its theoretical development can be deepened.

Material and methods:

The method used fits the needs of the study and its objectives, although it presents some errors of form. For example, in line 137, Chi square appears in capital letters; it should be corrected by writing it correctly.

Results

-It is recommended that in Table 1 the significance of the p values be indicated with asterisks (*,**) with the corresponding degree, not only marking it in bold.

-The significance level for each variable should be specified in Table 2.

Ditto for the rest of the tables

-The lack of text between the end of the results section and the discussion section should be corrected.

-It is strongly recommended to perform a multivariate statistical analysis, e.g., correspondence analysis or principal component analysis to see how the different variables are related.

Discussion

-Although the discussion is presented logically and coherently, some references should be introduced in the first paragraphs.

Conclusions

-Conclusions need to be developed.

References:

References are correct and fit the interest of the work and the stated objectives.

Author Response

STATUS: Pending minor revisions

Point 1: Summary: Although the summary is correct, the mean and deviation of the age of the participants should be included.

Response 1: Corrected after suggestion. All changes are highlighted in yellow in the revised manuscript.

Introduction and theoretical framework: The theoretical framework presents the main orientations and references so that the theoretical background of the paper can be presented and its theoretical development can be deepened.

Point 2: Material and methods: The method used fits the needs of the study and its objectives, although it presents some errors of form. For example, in line 137, Chi square appears in capital letters; it should be corrected by writing it correctly.

Response 2: We apologize for the mistake, indeed we wrote to the journal and the Chi square was removed although you have seen the previous version. Moreover, we have reviewed this section and some mistakes have been corrected. Thank you.

Point 3: Results: It is recommended that in Table 1 the significance of the p values be indicated with asterisks (*,**) with the corresponding degree, not only marking it in bold.

Response 3: As the P value is given in the Table, the reader can see the degree in that column; thus, in our honest opinion, to include those asterisks is not needed since it duplicates information.

Point 4: -The significance level for each variable should be specified in Table 2.

Response 4: P values have been added.

Point 5: Ditto for the rest of the tables

Response 5: P values have been added in all Tables.

Point 6: The lack of text between the end of the results section and the discussion section should be corrected.

Response 6: Corrected after suggestion.

Point 7: It is strongly recommended to perform a multivariate statistical analysis, e.g., correspondence analysis or principal component analysis to see how the different variables are related.

Response 7: Thank you for the kind suggestion. Following your proposal, we have conducted a principal component analysis using parallel analyses to determine how the variables are related. The analysis gave us 4 clusters that are shown in the newly added figure A1. In summary, the analyses created a cluster for body composition, a cluster for the physical tests, a cluster for moderate to vigorous physical activity and a cluster with light physical activity and inactivity.

Given the results, we decided to change the linear regression analyses (step-wise method) including each cluster as a specific model instead of the model used previously in the first draft (Table 4). Nevertheless, the final model remained the same. All changes have been highlighted in yellow.

Point 8: Discussion: Although the discussion is presented logically and coherently, some references should be introduced in the first paragraphs.

Response 8: We have included more references according to the reviewer’s suggestion.       

Point 9: Conclusions need to be developed.

Response 9: This section has been thoroughly detailed.

Point 10: References are correct and fit the interest of the work and the stated objectives.

Response 10: Thank you for your time and your valuable comments to improve our manuscript.